# Recent Progress of Atomic Magnetometers for Geomagnetic Applications

**DOI:** 10.3390/s23115318

**Published:** 2023-06-03

**Authors:** Yuantian Lu, Tian Zhao, Wanhua Zhu, Leisong Liu, Xin Zhuang, Guangyou Fang, Xiaojuan Zhang

**Affiliations:** 1Aerospace Information Research Institute, Chinese Academy of Sciences, No. 9 Dengzhuang South Road, Beijing 100094, Chinagyfang@mail.ie.ac.cn (G.F.);; 2School of Electronic, Electrical and Communication Engineering, University of the Chinese Academy of Sciences, Beijing 100049, China

**Keywords:** atomic magnetometers, geomagnetic applications, technology trends

## Abstract

The atomic magnetometer is currently one of the most-sensitive sensors and plays an important role in applications for detecting weak magnetic fields. This review reports the recent progress of total-field atomic magnetometers that are one important ramification of such magnetometers, which can reach the technical level for engineering applications. The alkali-metal magnetometers, helium magnetometers, and coherent population-trapping magnetometers are included in this review. Besides, the technology trend of atomic magnetometers was analyzed for the purpose of providing a certain reference for developing the technologies in such magnetometers and for exploring their applications.

## 1. Introduction

The precise measurement of weak magnetic fields is the key to solving problems in many basic and applied disciplines, such as geophysics [1,2], archeology [3,4], cosmophysics [5,6], and biophysics [7]. Common weak magnetic sensors include atomic magnetometers [8], induction magnetometers [9,10], fluxgate magnetometers [11], proton magnetometers [12], superconducting quantum interference devices [13,14], anisotropic magnetoresistance magnetometers [15], planar Hall effect magnetometers [16], giant magneto-impedance magnetometers [17], etc. Due to their different characteristics, various weak magnetic sensors are widely used in different fields. Among them, the atomic magnetometers are widely used in geophysics due to their unique features, including the high sensitivity, medium volume, anti-vibration ability, and uncooled working condition.

Pieter Zeeman discovered that a spectral line of atoms could be split into several components in the presence of an external magnetic field in 1896, known as the Zeeman effect [18]. The degree of splitting is a function of the field strength, so the effect can be used to measure the magnetic field. Neglecting the effect from the high-order magnetic field term under the geomagnetic field, the distance of splitting lines is linearly proportional to the strength of the external magnetic field. In the early 1950s, Alfred Kastler proposed and developed the optical pumping technology based on the Zeeman effect [19,20]. The ground-state atomic ensemble in a uniform distribution was redistributed by the polarized light emitted from the electrodeless discharge lamps or lasers, which led to the spin polarization and generated a macroscopic magnetism. Consequently, the combination of the optical resonance and the magnetic resonance is a key to the atomic magnetometers. In 1957, H. G. Dehmelt firstly proposed that the magnetic field can be measured by detecting the precession of sodium atoms, which is regarded as the beginning of optically pumped magnetometers [21]. In September of the same year, Bell and Bloom, two doctors from Varian Associates, realized the optically pumped magnetometer in experiments and developed the synchronous optical pumping technology at a later time [22,23]. Since then, the atomic magnetometer technology based on alkali-metal atoms and helium atoms has rapidly developed at almost the same time [24,25,26,27].

The spin exchange relaxation phenomenon, existing in the traditional optically pumped magnetometers, enlarges the optical resonance linewidth. This limits the sensitivity of traditional atomic magnetometers to some extent. In the 1970s, Happer et al. discovered that the spin exchange relaxation can be suppressed when the spin exchange rate is much greater than the Larmor precession frequency and gave a related theoretical explanation [28,29]. In 2002, Romalis’s group at Princeton University realized the spin exchange relaxation free (SERF) state of atoms and successfully developed the SERF atomic magnetometer for the first time, which improved the sensitivity of the atomic magnetic sensor to the fT level at low frequencies [30]. Although the SERF atomic magnetometer suppresses the influence of spin exchange relaxation and is currently one of the most-sensitive magnetometers [31], the dynamic range of the SERF atomic magnetometer is quite small, typically on the order of nanoteslas (nT). Although its dynamic range can be expanded through methods such as closed-loop control [32], its application in the geomagnetic field environment is still limited.

This review first briefly introduces the development history, composition, and classification of the atomic magnetometers, then reports the recent progress of total-field atomic magnetometers that reach the technical level of engineering applications, mainly including alkali-metal magnetometers, helium magnetometers, and coherent population-trapping (CPT) magnetometers. Finally, the technology trends of atomic magnetometers are presented.

## 2. Basic Components and Classification

### 2.1. Basic Components

Figure 1 is a schematic diagram of the components of an atomic magnetometer, which mainly includes three basic elements: a light source, a vapor cell, and a photodetector. Among them, the light source is used to change and detect the spin state of the atomic ensemble in the gas cell. Parameters such as the wavelength and polarization of the light are determined based on the atomic type and detection mechanism. A commonly used light source consists of a narrow-linewidth semiconductor laser [33], a low-noise electrodeless discharge lamp [34], and a low-power vertical-cavity surface-emitting laser (VCSEL) [35]. The vapor cell is the core unit of the atomic magnetometer, and the composition parameters of the gas chamber are determined and optimized based on specific experimental principles. The average magnetic moment of the atomic ensemble processes around the non-coaxial external magnetic field in the cell. The inner-wall-processing technology of cells and the cell parameters affect the accuracy and sensitivity of the atomic magnetometer. To reduce the wall collision relaxation, the cell is generally filled with a buffer gas or has an antirelaxation surface coating [36,37]. The photodetector detects the light absorption or the polarization plane rotation information after the interaction between the light and atoms, which can extract the magnetic field value from the Larmor precession frequency. The other components of the atomic magnetometer, such as the optical ones and coils, are selected based on the specific principles or the detection schemes for the magnetic measurement.

Based on the Zeeman effect, the atomic magnetometer uses different detection methods to measure the external magnetic field information. Atomic energy levels can split into fine and hyperfine structures due to the interaction between the internal angular momentaof the atom. Under the action of a weak magnetic field B, the hyperfine energy level with angular momentum quantum number *F* will split into 2F+1 Zeeman sub-levels, and the energy level intervals between the Zeeman sub-levels are equal. The energy difference ΔE between the adjacent Zeeman sub-levels is shown as [38]:(1)$$\Delta E=g_{F}\mu_{B}B,$$
where $g_{F}$ is the ground-state Lande’s factor and $\mu_{B}$ is the Bohr magneton. The total magnetic moment of the atom undergoes Larmor precession around the direction of the external magnetic field, and the precession frequency is ω0=ΔEΔEℏℏ=γB; γ is the gyromagnetic ratio of the atom.

### 2.2. Classification

With the development of atomic magnetometry, various types of devices have been developed for many application scenarios. At the same time, there are also many other classification methods. For example, based on the measured magnetic field parameters, magnetometers can be divided into scalar and vector ones [39,40,41]. According to the modulation method, the atomic magnetometers can be divided into magnetic-field-modulation magnetometers and all-optical ones [42,43]. Among them, magnetometers that can work under geomagnetic conditions mainly include the alkali-metal- or helium-based optically pumped magnetometers and the coherent population-trapping (CPT) magnetometers [44,45,46]. The magnetometers that work in the near-zero field are mainly referred to as SERF magnetometers and nonlinear magneto-optical rotation (NMOR) magnetometers [47,48,49].

The working principle of an optical pumping magnetometer mainly includes three processes: optical pumping, magnetic resonance, and optical detection. Firstly, the unpolarized atoms in a weak magnetic field exhibit a Boltzmann distribution on the Zeeman sub-levels, and the energy level intervals between adjacent sub-levels are proportional to the strength of the magnetic field. After absorbing circularly polarized light with a specific wavelength, the atoms are unevenly distributed in the ground-state sub-levels. The atomic ensemble exhibits a macroscopic magnetic moment, and the interaction between the magnetization vector M and the external magnetic field B can be described by the Bloch equation [50]:(2)$$\frac{dM}{dt}=M\times \gamma B+\frac{M_{0}-M}{T},$$
where $M_{0}$ is the equilibrium magnetization and *T* is the relaxation time, which is the characteristic time it takes for the magnetization of the system to return to its equilibrium state after being perturbed.

Then, the atoms are excited by the radio frequency (RF) magnetic field with a frequency of ω1=ΔEΔEℏℏ, and the atomic population of the ground-state Zeeman sub-levels changes. The atoms conforming to the transition rules continue to absorb the circularly polarized light, and the light intensity passing through the cell decreases. When the RF frequency $\omega_{1}$ is equal to the Larmor precession frequency $\omega_{0}$, the atomic ensemble generates magnetic resonance, and the light intensity reaches the minimum. The magnetic resonance frequency can be obtained by monitoring the light intensity, and then, the external magnetic field can be calculated through the gyromagnetic ratio. For the convenience of calculation, a rotational coordinate system X${}^{\prime }$Y${}^{\prime }$Z is introduced to rotate around the Z-axis at angular velocity $\omega_{1}$. The projections of the transverse components of the magnetization M on the X${}^{\prime }$ and Y${}^{\prime }$ axes are defined as *u* and *v*, respectively. In a rotating coordinate system, the steady-state solution of the Bloch equation is given as [51]:(3)$$\begin{array}{cc} \; & u=M_{0}\frac{\gamma B_{1}\Delta \omega T_{2}{}^{2}}{1+{\left(\Delta \omega T_{2}\right)}^{2}+{\left(\gamma B_{1}\right)}^{2}T_{1}T_{2}}\\ \; & v=-M_{0}\frac{\gamma B_{1}T_{2}{}^{2}}{1+{\left(\Delta \omega T_{2}\right)}^{2}+{\left(\gamma B_{1}\right)}^{2}T_{1}T_{2}}\\ \; & M_{z}=M_{0}\frac{1+{\left(\Delta \omega T_{2}\right)}^{2}}{1+{\left(\Delta \omega T_{2}\right)}^{2}+{\left(\gamma B_{1}\right)}^{2}T_{1}T_{2}} \end{array},$$
where $\Delta \omega =\omega_{0}-\omega_{1}$ is the detuning frequency of the rotational field and $B_{1}$ is the magnetic field strength of the RF field.

According to the different magnetic resonance signal components used for detection, the optically pumped magnetometers can be divided into Mx and Mz types [52]. The atomic magnetometer of the Mx type utilizes the transverse component of the magnetic moment, which is determined by the variables *u* and *v*, and the Mz magnetometer utilizes the longitudinal component of the magnetic moment that is described by the variable Mz. The direction of the beam and magnetic field for the magnetometers of Mx and Mz types are also different, as shown in Figure 2. The beam of Mz magnetometers is parallel to the measured magnetic field, while both the pump light and probe light have a certain angle to the magnetic field. According to the processing method of the photodiode signal, Mx magnetometers can recover the signals by using self-oscillating mode and phase-locked loop (PLL) mode [53,54]. The detected signal from the self-oscillating magnetometer is amplified and fed back to the RF coil, which makes those magnetometers have the characteristics of a fast response speed and simple structure. The PLL-mode magnetometer locks the Larmor frequency in a closed loop through the phase-locked loop technology. Compared with the self-oscillating magnetometer, the PLL one has a lower response speed, but a higher accuracy of measurement is more suitable for the magnetic field observation with the absolute measurement. In practical applications, the specific detection schemes are usually selected based on the specific application scenarios.

Coherent population-trapping (CPT) refers to the phenomenon when the atoms in the ground-state undergo simultaneous transitions through different channels, causing mutual interference between the transition channels, so that the atoms are trapped in the superposition state with the ground-state, and no transition occurs. The CPT magnetometer uses the signal generated by the CPT effect to measure the magnetic field [55,56]. As shown in Figure 3, taking the D${}_{1}$ line of the ${}^{87}$Rb atom as an example, there are three groups of “Λ”-type three-level subsystems. In the experiment, two coherent beams of light are prepared by modulating the monochromatic light with high-frequency microwave, and three groups of CPT resonance processes occur in the atomic ensemble. Atoms are trapped in the ground-state and will not absorb the laser with the corresponding frequency, so there are three transmission peaks in the transmitted light. Usually, the signal of enhanced transmission caused by the CPT effect is called the electromagnetically induced transparency (EIT) signal. The gap between adjacent transmission peaks is proportional to the strength of the external magnetic field, so the magnitude of the magnetic field can be reserved by measuring the frequency interval.

The SERF atomic magnetometer operates in a near-zero magnetic field because the implementation of SERF states requires the conditions of high atomic density and a low magnetic field. Its composition is shown in Figure 4. The basic working principle is that the resonant circularly polarized light is used to pump the alkali-metal atom vapor and the magnetic field induces the polarized atomic spin precession, then the atomic spin is detected through linearly polarized detection light close to the resonant frequency; thereby, the magnetic field measurement is subsequently realized. Due to the interaction of the external magnetic field, pumping light, and various relaxation mechanisms in the cell, the equation of evolution for the atomic ground-state can be described by the Bloch equation [57]:(4)$$\frac{dS}{dt}=\frac{1}{q}\left[\gamma_{e}B\times S+R_{\mathrm{op}}\left(s\cdot \hat{z}-S\right)-R_{\mathrm{rel}}S\right],$$
where S is the alkali-metal electron spin and *q* is the slowing-down factor, indicating the extent to which the hyperfine interaction reduces the spin precession frequency [58]. $\gamma_{e}$ is the gyromagnetic ratio of a bare electron, B is the magnetic field, $R_{\mathrm{op}}$ is the optical pumping rate, *s* is the average photon spin, and $R_{\mathrm{rel}}$ is the atomic spin relaxation rate, which arises from spin destruction collisions, spin exchange collisions, wall collisions, magnetic field gradients, and so on. The steady-state solution along the direction of the detecting light can be simplified as:(5)$$S_{x}=S_{0}\frac{\gamma_{e}}{R_{op}+R_{rel}}B_{y},$$
where $S_{0}$ is the equilibrium spin polarization. According to Equation (Equation 5), the SERF magnetometer generally measures the vector information of the magnetic field perpendicular to the plane that is formed by the pump and detection light beams.

## 3. The State of Development and Application

Based on the above classification, the application and development state of the atomic magnetometers that can work in the geomagnetic environment will be separately discussed for the alkali-metal atomic magnetometers, the helium optically pumped magnetometers, and the CPT magnetometers.

The specifications of the magnetometers should meet the application requirement. The range of the Earth’s surface magnetic field is approximately 25 μT to 65 μT, so the dynamic range of a magnetometer should encompass this range of magnetic fields [59]. Geomagnetic measurements typically serve two applications: absolute value measurement, such as geomagnetic observation at magnetic observatories or satellites, and magnetic anomaly detection, including geological exploration, submarine detection, unexploded object detection, and more. Absolute accuracy and long-term stability are essential for measuring absolute values. Since magnetic anomalies are often on the scale of pT or even smaller, sensitivity is the crucial factor in detecting the smallest deviations [60]. For motion platform applications such as airborne or maritime systems, it is important to carefully investigate heading errors and dead zones to prevent null output or “striping” in the final results. Gradient tolerance is a crucial criterion for detection applications involving significant magnetic anomalies, such as inspections of smelting sites, submerged pipelines, and unexploded ordnance.

### 3.1. Alkali-Metal Atomic Magnetometers

The optically pumped magnetometer has the characteristics of high measurement accuracy, high sensitivity, and small volume. It can measure both the total and gradient of the magnetic field, which is widely used in the measurement of space magnetic fields [6], geophysical exploration [61], military defense [62], etc. [63]. The earliest technology of atomic magnetometers was based on alkali-metals, and the potassium, rubidium, and cesium atoms were the most-often used atomic sources for such techniques. Considering the structure of the energy level, the shift of the Zeeman frequency from potassium is small and the intervals between the atomic spectra are quiet large, so the theoretical accuracy and sensitivity of potassium magnetometers are higher than those of the rubidium and cesium ones [64]. However, the technical maturity of cesium and rubidium magnetometers is greater than potassium magnetometers. The electrodeless discharge lamps were used in alkali-metal-based magnetometers as light sources in early times [24]. In recent years, various new systems of magnetometers based on laser sources have emerged one after the other with the development of laser technology [34,65,66].

Alkali-metal-based atomic magnetometers that are used in proof-of-principle experiments pursue much higher sensitivity for a wider range of applications. S. Groeger et al. at the University of Friborg in Switzerland achieved an inherent sensitivity of 15 ${\mathrm{fT}/\mathrm{Hz}}^{1/2}@$ 1 Hz in 2005 by optimizing the parameters of the laser cesium magnetometer [34]. Besides, the team used the cesium magnetometer to map the cardiomagnetic fields of human beings, which are usually measured by SQUIDs [63]. In 2017, Weimin Sun’s group from Harbin Engineering University in China proposed an all-optical vector cesium magnetometer, which is based on the Bell–Bloom principle and achieves an amplitude sensitivity of 80 ${\mathrm{fT}/\mathrm{Hz}}^{1/2}$ and a directional sensitivity of 0.1${}^{\circ }/{\mathrm{Hz}}^{1/2}$. The vector magnetometer is a suitable candidate for positioning and navigating applications [67]. In 2021, Qiang Lin’s group at Zhejiang University measured the magnetic cobalt particles with an elliptical-light-pumped Mx rubidium magnetometer, as shown in Figure 5. A sensitivity of 0.2 ${\mathrm{pT}/\mathrm{Hz}}^{1/2}$ at 70 ${}^{\circ }$C was achieved, and this paved the way for the atomic magnetometers to measure the magnetic particles in biological and industrial applications [68].

In addition to pursuing a higher sensitivity, the miniaturization of optically pumped magnetometers has also been a hot research topic for a long time. In 2007, Peter D. D.Schwindt et al. at the National Institute of Standards and Technology (NIST) in the United States created a Mx chip-scale atomic magnetometer based on the experience of miniaturized atomic clocks, as shown in Figure 6. The magnetometer adopted VCSEL and a microfabricated Rb vapor cell of 1×2×1 ${\mathrm{mm}}^{3}$ to achieve a noise level of 5 ${\mathrm{pT}/\mathrm{Hz}}^{1/2}$@ 1–100 Hz [69]. In 2016, Haje Korth et al. at Johns Hopkins University developed a miniature scalar atomic magnetometer based on ${}^{87}$Rb atoms, which consists of a 1 ${\mathrm{mm}}^{3}$ cell and customized integrated circuits. The total mass of the magnetometer was less than 500 g, and the power consumption was less than 1 W. While possessing miniaturization characteristics, the sensitivity of the magnetometer did not decrease too much, and a noise level of 15 ${\mathrm{pT}/\mathrm{Hz}}^{1/2}@$ 1 Hz [70] was achieved. In 2020, G. Oelsner et al. from Leibniz Institute of Photonic Technology in Germany proposed a portable cesium magnetometer based on the light-shift-dispersed Mz mode, which can achieve a sensitivity of 140 ${\mathrm{fT}/\mathrm{Hz}}^{1/2}$ under the geomagnetic field [44]. In 2021, Yoel Sebbag et al. from the Hebrew University of Jerusalem in Israel demonstrated a rubidium magnetometer with chip-scale nanophotonic components. The cores of this magnetometer were a 2mm×2mm×30 μm cell and an integrated photonic spin selector. It realized a micrometer-scale spatial resolution and a sensitivity of 700 ${\mathrm{pT}/\mathrm{Hz}}^{1/2}$ [71].

Presently, the companies that manufacture the products of the traditional alkali-metal optically pumped magnetometer are primarily Geometrics, GEM, Scintrex, and CAE. The products from Geometrics in the United States include marine magnetometers, land magnetometers, and airborne magnetometers [72]. Figure 7a is a photo of the airborne G-824A cesium magnetometer, and Figure 7b is a noise spectral plot of the G-824A. Its sensitivity is 0.6 ${\mathrm{pT}/\mathrm{Hz}}^{1/2}$, with an absolute accuracy less than 3 nT, and the directional error is ±0.15 nT. The high performance and reliability of this system make it available for mapping the geologic structures or for mineral exploration. The typical products from Scintrex in Canada include CS-3 and CS-VL high-precision cesium magnetometers, possessing a sensitivity of 0.6 ${\mathrm{pT}/\mathrm{Hz}}^{1/2}$ and an absolute accuracy higher than 2.5 nT. The CAE Company in Canada mainly provides cesium magnetometers, most of which are mainly used for military applications. Currently, it has delivered more than 2000 magnetic anomaly detection (MAD) systems to military forces around the world. The newest equipment from CAE is MAD-XR, equipped with a miniaturized cesium optically pumped magnetometer, which has been successfully installed on the U.S. Navy’s MH-60R Seahawk helicopter [73]. The GEM Company in Canada mainly provides potassium magnetometers and proton magnetometers. The typical product is the GSMP-35 series potassium magnetometers, with a magnetic measurement range of 15,000 nT to 120,000 nT, an absolute accuracy of ±0.1 nT, and a sensitivity of 0.3${\mathrm{pT}/\mathrm{Hz}}^{1/2}$@1Hz [74]. Due to its high accuracy and sensitivity, the potassium magnetometer can be used in geomagnetic observatories. E. Pulz at GeoForschungsZentrum Potsdam designed a Cs-K tandem magnetometer, which combines the advantages of fast magnetic response from cesium magnetometers, as well as high long-term stability from potassium magnetometers. The tandem magnetometer achieved an absolute accuracy better than 0.1 nT and a long-term stability better than 20 pT/year [75].

With the rapid development of MEMS manufacturing technology, the performance of miniaturized alkali-metal magnetometers has gradually improved, and some commercial products have emerged in the market. The MFAM produced by Geometrics is a miniature cesium magnetometer with a sensitivity of 2 ${\mathrm{pT}/\mathrm{Hz}}^{1/2}$ and a sensor dimension of 33×25×32 ${\mathrm{mm}}^{3}$. As shown in Figure 8a, the two sensors allow for the individual or gradient measurements, operate without a dead zone, and can compensate for heading errors [76]. The dead zone of a magnetometer refers to a specific region or range where the magnetometer’s sensitivity is significantly reduced or compromised, leading to inaccurate or unreliable measurements. Heading errors refer to inaccuracies or deviations in the measurement of the magnetic field direction, typically in reference to the Earth’s magnetic field. As shown in Figure 8b, the total-field magnetometer QTFM Gen-2 produced by Quspin Company in the United States is a pulsed-laser-pumped rubidium magnetometer. The QTFM Gen-2 is based on the optical detection scheme of free induction decay (FID), which can measure the scalar and three-axis vectorial magnetic fields. FID refers to the phenomenon in which the magnetic field signal, generated by an external magnetic pulse or excitation, gradually diminishes over time without any further stimulation. By measuring the FID, characteristics such as relaxation times, resonant frequencies, or magnetic field strengths can be determined. The sensitivity of the QTFM Gen-2 in scalar mode is better than 3 ${\mathrm{pT}/\mathrm{Hz}}^{1/2}$, and the total weight of the sensor head and electronics is only 15 g [77]. An optical magnetic gradiometer (OMG) based on FID has also been developed by Twinleaf Company. The sensor has two cells, and the sensor volume is 16×36×136 ${\mathrm{mm}}^{3}$. The sensitivity of the OMG is better than 0.2 ${\mathrm{pT}/\mathrm{Hz}}^{1/2}$ [78]. Limes et al. at Twinleaf demonstrated the portable OMG for the detection of biomagnetism in natural environments [79]. As shown in Figure 9, the auditory-evoked fields were detected in Earth’s field, which showed the potential of the array detection of unshielded biomagnetism. Miniaturized commercial alkali-metal magnetometers have the characteristics of small size and low power consumption and have great application prospects in magnetic anomaly detection of unmanned aerial platforms and magnetocardiography and magnetoencephalography in magnetically unshielded environments.

### 3.2. Helium Optically Pumped Magnetometers

The principle of the helium optically pumped magnetometer is similar to that of the alkali-metal magnetometer, with an additional step of electromagnetic excitation [80]. High-frequency excitation is used to excite the helium in the cell from the ground-state $1^{1}S_{0}$ to the metastable state $2^{3}S_{1}$. The helium atoms in the metastable state are relatively active, and the magnetic field information can be measured through optical pumping and magnetic resonance. Since the gyromagnetic ratio of helium atoms is several times larger than that of alkali-metals, the theoretical sensitivity of helium magnetometers is higher than that of alkali-metal atomic magnetometers. Meanwhile, the metastable energy level structure of helium atoms is simple, and the nuclear magnetic moment is zero, which has high absolute measurement accuracy. Therefore, helium magnetometers have many important applications in the field of geomagnetic observation [43,81,82].

The optical pumping of ${}^{4}$He was first discovered by Colegrove, F. D. et al. [26], and A. Keyser et al. developed the first helium magnetometer for measurements in geomagnetic applications [83]. Polatomic of the United States, CEA-Leti of France, Peking University of China, and China Geological Survey are the primary research institutions and suppliers of helium magnetometers at the moment. Polatomic Inc. is one of the earliest organizations engaged in the research of helium magnetometers. The company has in-depth cooperation with American government and has customized a series of helium magnetometers for airborne, underwater, and space applications. For example, the American P-3C Orion anti-submarine patrol aircraft is equipped with the AN/ASQ-233 helium magnetometers produced by Polatomic [84]. The Cassini-Huygens spacecraft launched by NASA in 1997 was equipped with a vector helium magnetometer with scalar modes to measure the magnetic field and electromagnetic interaction of Saturn and Titan. The magnetometer was jointly developed by NASA, the Jet Propulsion Laboratory (JPL), and the Polatomic, with a sensitivity of 5 ${\mathrm{pT}/\mathrm{Hz}}^{1/2}$ in vector mode [85]. A set of scalar magnetometers provided was installed on the SAC-C magnetic measurement satellite launched in 2000. The magnetometer uses a helium lamp to pump two cells arranged in an orthogonal arrangement and is used to monitor the long-term changes in the Earth’s magnetic field [86].

In addition, B. Chéron’s group in France studied various schemes of helium magnetometers, including laser-intensity-modulated magnetometers, laser-frequency-modulated magnetometers, etc. [87,88,89]. In 2013, the European Space Agency (ESA) launched the SWARM satellite, which is equipped with an absolute helium laser optically pumped magnetometer developed by the CEA-Leti of France, as shown in Figure 10a. It can measure scalar and vector magnetic fields with resolutions of 1 ${\mathrm{pT}/\mathrm{Hz}}^{1/2}$ and 1 ${\mathrm{nT}/\mathrm{Hz}}^{1/2}$, respectively [90]. The geomagnetic maps acquired with ${}^{4}$He magnetometers on board the SWARM satellites are shown in Figure 10b [5]. The space magnetometer can simultaneously measure scalar and vector information, which provides geomagnetic field modeling studies with different input datasets. In 2021, a ${}^{4}$He vector zero-field optically pumped magnetometer based on parametric resonance was studied by CEA-Leti [40]. The magnetometer can work in the geomagnetic field with a real-time field compensation. As shown in Figure 11, the magnetometer in closed-loop achieved an intrinsic sensitivity of 130 ${\mathrm{fT}/\mathrm{Hz}}^{1/2}$, which is an advantageous alternative to fluxgates. The China Geological Survey has been conducting research on helium lamp optically pumped magnetometers since the 1970s. The representative products mainly include the HC-2000K and HC-90K aviation helium optically pumped magnetometers [91]. In 2020, Hong Guo’s group from Peking University presented an all-optical self-oscillating ${}^{4}$He atomic magnetometer, achieving a sensitivity of 1.7 ${\mathrm{pT}/\mathrm{Hz}}^{1/2}$. A liquid crystal to compensate for optical phase shift was used to achieve self-oscillation. Compared to the self-oscillating cesium atomic magnetometer, this magnetometer has greater gradient tolerance [92].

### 3.3. CPT Magnetometers

The CPT magnetometer measures the frequency interval of the EIT peaks when CPT resonance occurs and converts the frequency into magnetic field information by using the gyromagnetic ratio and other coefficients. Compared with the fluxgate magnetometer, the CPT magnetometer measures the absolute value of the magnetic field based on the direct detection of the Larmor precession and has a smaller long-term drift [93]. Compared with the optically pumped magnetometer, the CPT magnetometer has no dead zone and a wide dynamic range (from 100 nT to 100,000 nT) [94]. The scalar CPT magnetometer is an all-optical atomic magnetometer that does not require RF field excitation, so it has little external electromagnetic interference. There is no mutual interference between the sensors in the proximity.

In 2004, Peter D. D. Schwindt et al. from NIST developed a chip-scale CPT magnetometer using a 1 ${\mathrm{mm}}^{3}$ microfabricated rubidium vapor cell, as shown in Figure 12. The magnetometer consumes 195 mW of power and achieves a sensitivity of 50 ${\mathrm{pT}/\mathrm{Hz}}^{1/2}$@10Hz [95]. In 2010, the Austrian Academy of Sciences developed a Coupled Dark State Magnetometer (CDSM) based on the CPT principle [96]. CDSM uses several parallel coupled dark state resonances to measure external magnetic fields, reducing interference effects such as optical shift, high-order magnetic-field-induced shift, and temperature-dependent shift. In 2021, Yanying Feng’s group from Tsinghua University developed a CPT magnetometer with microfabricated vapor cells. The closed-loop magnetometer achieved a sensitivity of 210.5 ${\mathrm{pT}/\mathrm{Hz}}^{1/2}$@1Hz. In recent years, the CPT atomic magnetometer scheme for measuring in the vector magnetic field and measuring in the near-zero field has also been proposed and verified [35,97].

In terms of applications, as shown in Figure 13, the CDSM developed by the Austrian Academy of Sciences was successfully carried aboard the China Seismo-Electromagnetic Satellite (Zhang Heng-1 01 satellite) in 2018 for investigating natural electromagnetic phenomena. The accuracy of the magnetometer is 0.19 nT, and the noise level is less than 50 ${\mathrm{pTrms}/\mathrm{Hz}}^{1/2}$ [98]. In 2023, CDSM will be aboard the Jupiter Ice Moons Explorer launched by the ESA. It will study the magnetic field around Jupiter’s moons, achieving extraterrestrial magnetic field modeling and space resource exploration [99].

## 4. Technology Trends

As mentioned above, atomic magnetometers have become indispensable sensors in many application fields after the technological development over the decades. However, the development of application has also put forward increasingly more requirements for the performance of atomic magnetometers, such as sensitivity, absolute precision, power consumption, volume, etc. Referring to the existing literature, it is believed that there are two main technical technology trends in atomic magnetometers’ development.

### 4.1. Ultra-High-Sensitivity Atomic Magnetometer Technology

For the sake of the requirements for detecting weak magnetic signals in geomagnetic environments, such as the detection of biomagnetism in unshielded environments [79], ultra-distant magnetic communication at low frequency [100], unexploded ordnance detection (UXO) on small targets [101], etc., the sensitivity and accuracy of atomic magnetometers operating in geomagnetic environments need to be further improved. On the one hand, the performance of existing total-field atomic magnetometers can be further optimized. From the perspective of key device performance, a lower-noise light source can be designed [102]; new processes have been introduced to enhance the lifetime and anti-relaxation performance of the cell [103], thereby approaching the theoretical sensitivity of atomic magnetometers. The detection strategy can be improved from the standpoint of increasing the signal-to-noise ratio. For example, hybrid atom vapor cells such as Cs-K and Cs-${}^{4}$He are used to combine the advantages of various atomic magnetometers [104,105]. In addition, the gradiometer structure based on FID can better suppress dead zones and common mode noise [106]. On the other hand, it is necessary to continuously research new magnetic measurement principles. For example, by using magnetic compensation coils or parametric modulation methods, the SERF atomic magnetometer and the NMOR magnetometer, which have extremely high theoretical sensitivity in zero field, have been extended to geomagnetic applications [79,107]. Besides, the technology of spin compression can be used to break through the fundamental quantum noise limits [108].

### 4.2. Chip-Scale Atomic Magnetometer Technology

For the purposes of unmanned platform magnetic detection or biomagnetism detection, such as unmanned aerial vehicle magnetic detection [109], magnetoencephalography [110], underwater magnetic environment monitoring [111], etc., it is necessary to develop low-power and miniaturized chip-scale atomic magnetometers. Currently, most miniaturized magnetometers are based on MEMS cells and VCSEL to reduce the size and power consumption of atomic magnetometers, such as the commercial magnetometers from Quspin and Twinleaf [77,78]. The further integration of key components such as cells is a very important research topic. For example, replacing space optical devices with nanostructures such as optical waveguides and hollow fibers is an important part of realizing chip-scale atomic magnetometers [112]. On the other hand, the mechanism of light–atom interaction influenced by nanostructures needs to be further studied to improve the sensitivity of chip-scale atomic magnetometers [113]. In addition, the chip-scale atomic magnetometers are expected to form composite sensors with other chip-scale sensors to achieve multi-parameter and multifunctional measurement [114], and they can be applied in emerging fields such as biomedicine, magnetic mapping, and basic physics research.

## 5. Conclusions

In summary, the development history, basic working principles, and classifications of atomic magnetometers were briefly presented and discussed. Focusing on the engineering applications of magnetometers, the recent progress of typical total-field atomic magnetometers was reported, mainly including the optically pumped magnetometers and the CPT magnetometers. Finally, the future technical development trends of atomic magnetometers were analyzed, by which we hope to induce some inspiration for the research on atomic magnetometers and their applications.

## Figures and Tables

**Figure 1 sensors-23-05318-f001:**
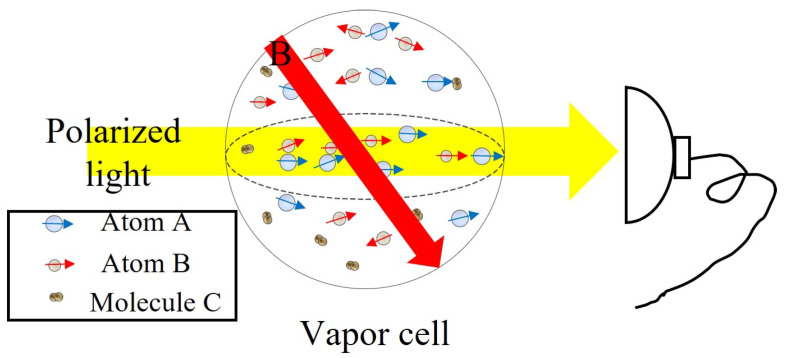
Schematic diagram of the basic components of an atomic magnetometer.

**Figure 2 sensors-23-05318-f002:**
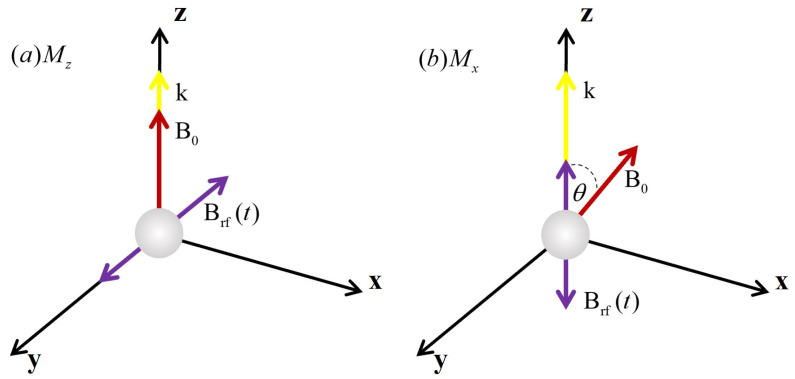
Schematic diagram of the direction between beam k and magnetic field $B_{0}$ of Mz-type and Mx-type magnetometers.

**Figure 3 sensors-23-05318-f003:**
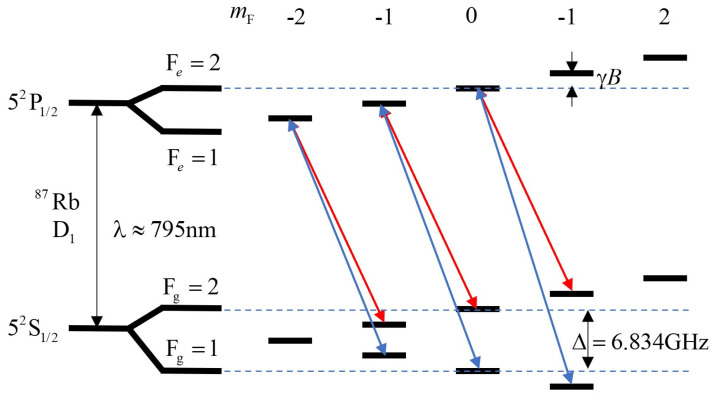
Scheme of Λ-type three-level subsystem in D${}_{1}$ line hyperfine structure of ${}^{87}$Rb.

**Figure 4 sensors-23-05318-f004:**
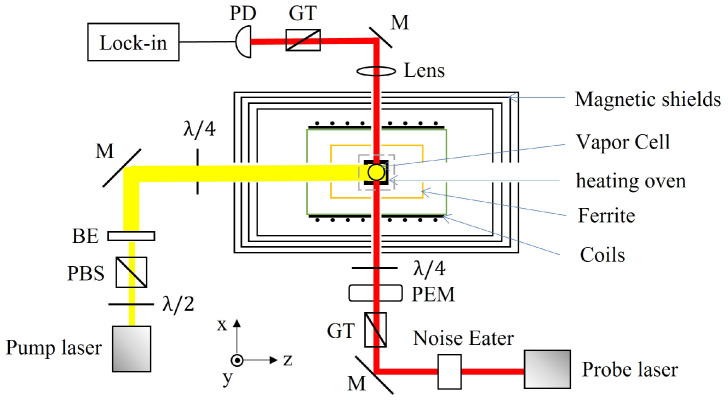
Schematic diagram of the SERF magnetometer.

**Figure 5 sensors-23-05318-f005:**
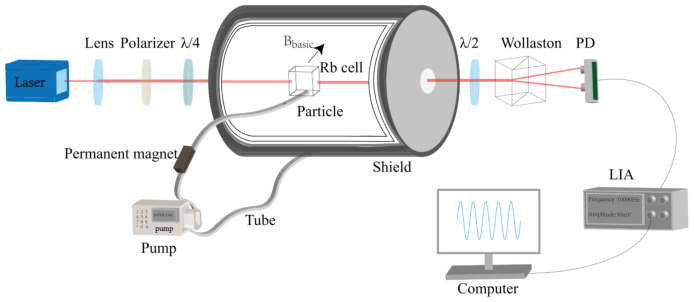
Schematic diagram of the elliptical-light-pumped Mx rubidium magnetometer designed by Qiang Lin’s group [68].

**Figure 6 sensors-23-05318-f006:**
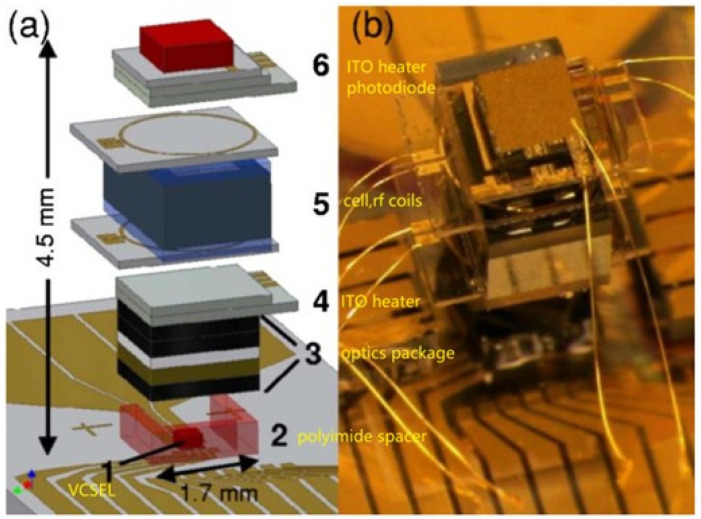
(**a**) Schematic of the chip-scale atomic magnetometer. The components are 1—VCSEL, 2—polyimide spacer, 3—optics package, 4—ITO heater, 5—cell with rf coils, 6—ITO heater and photodiode assembly. (**b**) Photograph of the magnetic sensor designed by NIST [69].

**Figure 7 sensors-23-05318-f007:**
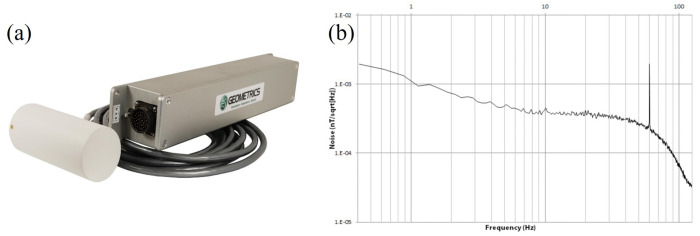
Photograph of cesium optically pumped magnetometer G-824A from Geometric Company (**a**) and noise spectral plot of G-824A (**b**) [72].

**Figure 8 sensors-23-05318-f008:**
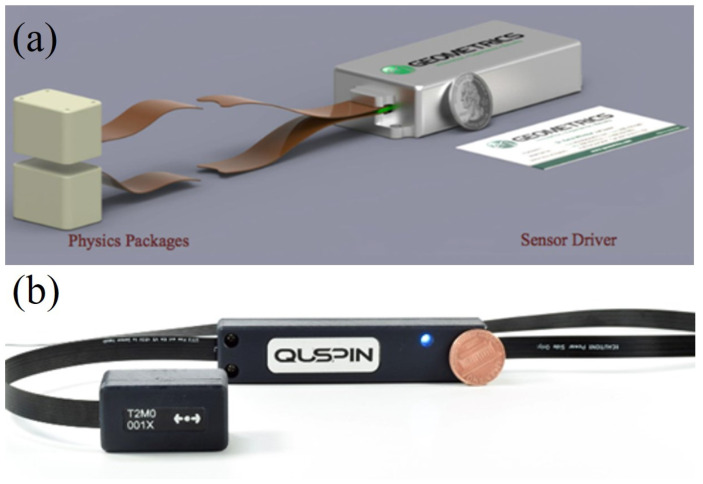
Product rendering of MFAM from Geometrics (**a**) [76] and QTFM Gen-2 from Quspin (**b**) [77].

**Figure 9 sensors-23-05318-f009:**
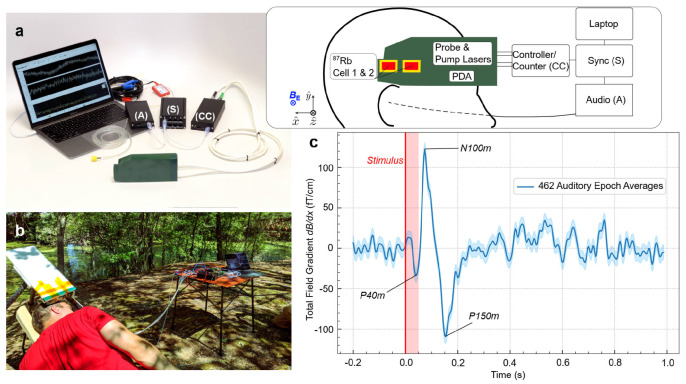
Auditory -evoked fields detected unshielded in Earth’s field (**c**) with a portable OMG designed by Limes et al. [79] (**a**,**b**).

**Figure 10 sensors-23-05318-f010:**
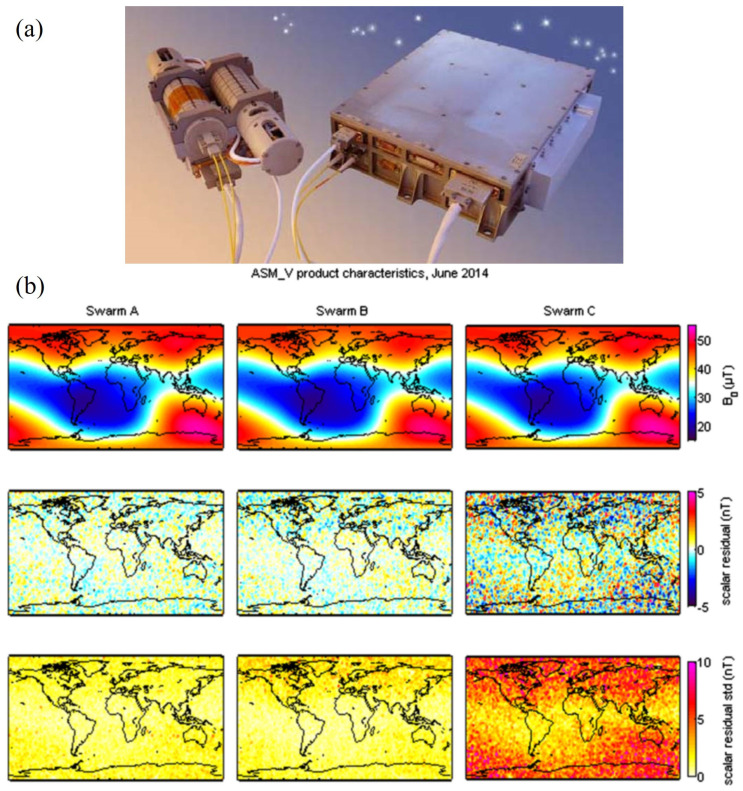
Schematic diagram of absolute scalar helium laser optically pumped magnetometer loaded on SWARM satellite (**a**) [90]. Geomagnetic maps measured by magnetometers on board the SWARM satellites (**b**) [5].

**Figure 11 sensors-23-05318-f011:**
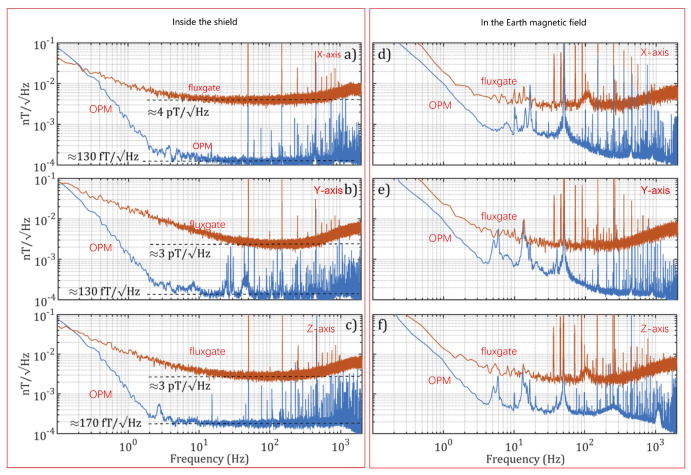
The sensitivity plots of the ^4^He vector optically pumped magnetometer in closed-loop designed by CEA-Leti [40]. Noise measurements inside the shield along the X (**a**), Y (**b**), and Z (**c**) axes. Noise measurements in the ambient Earth magnetic field along the X (**d**), Y (**e**), and Z (**f**) axes, OPM measurements in blue, and fluxgate ones in orange.

**Figure 12 sensors-23-05318-f012:**
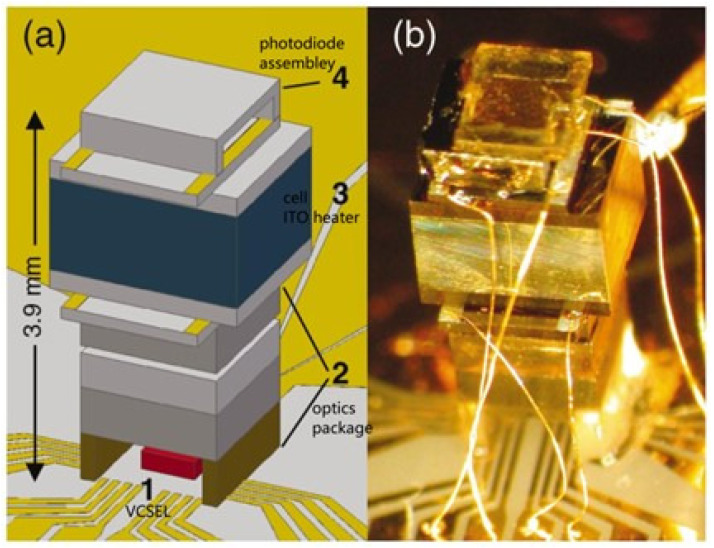
(**a**) Schematic of the chip-scale CPT atomic magnetometer. The components are 1—VCSEL, 2—optics package, 3—cell with ITO heaters, 4—photodiode assembly. (**b**) Photograph of the CPT atomic magnetometer designed by NIST [95].

**Figure 13 sensors-23-05318-f013:**
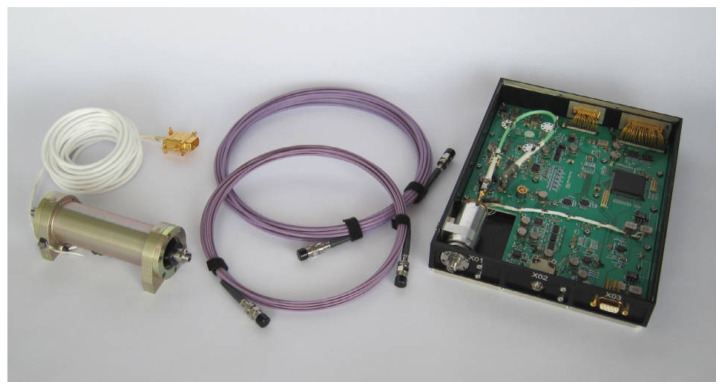
Photograph of coupled dark-state magnetometer developed by Austrian Academy of Sciences [98].

## Data Availability

Not applicable.

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
