# Peer review of "Recent Progress of Atomic Magnetometers for Geomagnetic Applications"

_sensors, 2023, doi:10.3390/s23115318_

Round 1

Reviewer 1 Report

The work is well organized has great level of details. 

What is missing the link between the high senistivity mangnetometers and the 

Geomagnetic Applications. I suggest you add a section or subsection talking about the requirements of  Geomagnetic Applications in terms of sensitivity, frequency stability, etc. 

also there is a typo in page 2 lines 43 and 44, it is "spin-exchange relaxation free" not "spin-free exchange relaxation"

Reviewer 2 Report

In this paper the authors present a review of recent atomic magnetometers, including their technological applications, with particular attention to geomagnetic applications. The topic is for sure of interest to a large community of scientists, the paper is  general well written (apart from minor issues I’ll tell later) and well documented. With some improvements, it deserves a publication. To be specific, in my opinion the paper can be improved under two aspects:

1) it is not clear what is the audience the paper is addressing. For an expert reader, it might be useful to have a paper resuming the most recent achievement in this field but probably at the very end the paper turns out to be almost useless. I think that a review should be useful to someone that wants to approach this topic, like a student for example. In this regard, the paper can be improved a lot. For example the theoretical part can be enriched in a number of ways, where do the relaxation terms come from? How much are they? Also, it would give more physical insight to report some graph of sensitivity, like that reported for example in PHYSICAL REVIEW APPLIED vol. 14, 064067 (2020). These are only example, actually the review can be extended in each section, otherwise as it is, it looks like a mere collection of different techniques.

2) The second main issue concerns the applications. The authors list a number of instruments embarked on board of space missions or for geophysics applications but it would be interesting to see some results obtained with these instruments, at least for a couple of them. This would make the paper more complete and more interesting to read. What kind of results have been obtained with this kind of instruments?  Are they important? Why?

Other issues:

1) in fig. 1 there are atoms with red and blue arrows but it is not clear what the colors are referring to.

2) line 165 and line 254; it is not explained what the slowing factor q and the dead zone are. This is an example of what I mean when I say that for people who know the topic the review is almost a mere collection of devices and for people who don’t know they should struggle and refer to the literature, whereas I would prefer the paper be self-consistent, at least to a certain extent.

3) gamma_e in eq (4) is the same as in line84 and in Eq 2? Please check in order to avoid confusion in the reader

3) line 257, again, the free induction decay is not described, scalar and vectorial magnetic field are just mentioned. Please look at the completeness of other reviews like Resonant nonlinear magneto-optical effects in atoms D. Budker, W. Gawlik, D. F. Kimball, S. M. Rochester, V. V. Yashchuk, and A. Weis

Rev. Mod. Phys. 74, 1153, and similar works.

Minor typo or language mistakes:

1) lines 72 and 112: Lamar to be changed in Larmor

2) line 40: This limits the sensitivity of traditional atomic magnetometers in an extant degree. Probably to be changed in This limits the sensitivity of traditional atomic magnetometers to some extent, please check

3) line 84 and line 112: the precession frequency is called omega in line 84 but in line 112 the larmor frequency is called omega0, so probably also the omega in line 84 should be changed on omega0

4) line 87-88, there some redundancy there,  the sentence can be changed as follows: With the development of the atomic magnetometry, various types of devices have been developed for many application scenarios.

5) line 104: externally can be removed

6) line 227 in the United States, please correct;

7) line 266, remove magnetic after the dot.

8) line 339; The noise to be correct in the noise

9) line 348, It is believed to be correct in it  is believed

10) line 353 ordnance probably is not an English word

Definitely, I think that it is interesting and useful to have such a review, provided the previous points are addressed.

The English is in general good. Some typo/mistakes still present

Reviewer 3 Report

In this manuscript, the authors report the recent progress of total-field atomic magnetometers. The references are detailed, and it can provide a good overview of the technology of atomic magnetometer. Besides, the manuscript exhibits the technology trend of total-field atomic magnetometers, which could provide a certain reference for developing the technologies and for exploring their applications. Therefore, I recommend to accept the paper after minor revision. The comments are as follows:

 1.     Please check the copyright rights to use all the figures.

2.     Page1, Line13. "Common weak magnetic sensors include atomic magnetometers[8], induction magnetometers[9,10], fluxgate magnetometers[11], proton magnetometers[12] and superconducting quantum interference devices[13,14], etc." The common weak magnetic sensors should also include TMR, AMR, GMR, GMI, et al, or authors can give the specific range of "weak magnetic field".

3.     Page 2, Line 46-48. "Although the SERF atomic magnetometer suppresses the influence of spin exchange relaxation and is currently one of the most sensitive magnetometers, the dynamic range of the SERF atomic magnetometer is quiet small. This limits its application in the geomagnetic field environment." “quiet” should be “quite”. As authors stated that dynamic range of the SERF atomic magnetometer is very small, the authors should give the quantitative value of range. Besides, there are some work reported on increasing the dynamic range of the SERF atomic magnetometer. I recommend authors referring to them.

4.     Page 7, Line214. "……optical pump magnetometer……"should be written in "……optically pumped magnetometer……". Page 7, Line227. "……optical pump magnetometer……",the same grammar error. Please check the manuscript carefully.

5.     Page 8, Line235. "……of potassium optical pump." should be written in "potassium magnetometers." or "potassium optically pumped magnetometers." The authors should check the syntax and expression errors throughout the manuscript.

The quality of English language can be further improved.
